# Efficacy of Jackfruit365™ Green Jackfruit Flour Fortified Diet on Pegfilgrastim to Prevent Chemotherapy-Induced Leukopenia, Irrespective of Tumor Type or Drugs Used—A Retrospective Study

**DOI:** 10.3390/biom10020218

**Published:** 2020-02-02

**Authors:** Thomas Varughese, James Joseph, Rejeesh Menon

**Affiliations:** 1Department of Oncology, Renai Medicity Hospital, Palarivattom, Cochin, Kerala, Pin 682025, India; 2God’s Own Food Solutions Pvt. Ltd., VII/219, Kaprassery, Nedumbassery, Kerala, Pin 683102, India; jamesjoseph@jackfruit365.com; 3Department of Medicine, Central Washington Hospital, Confluence Health, Wenacthee, Washington 98801, USA; Rejeesh.Menon@confluencehealth.org

**Keywords:** jackfruit, chemotherapy-induced leukopenia, chemotoxicity, pegfilgrastim, plant based compounds, natural compounds

## Abstract

Chemotherapy-Induced Leukopenia (CIL) is associated with increased mortality and economic burden on patients. This study was conducted to evaluate whether inclusion of green jackfruit flour in regular diet of those patients receiving chemotherapy, could prevent CIL. This was a retrospective study conducted among a group of patients undergoing chemotherapy for solid tumors at Renai Medicity Hospital, Palarivattom, Cochin, Kerala, India, since June 2018. The study group comprised of 50 consecutive subjects, who were supplemented with green jackfruit flour diet in their regular diet and further followed up prospectively. The control group was retrospective with 50 subjects prior to June 2018, with no diet supplements. Those who received less than three cycles were excluded from either arm. The mean age of the participants in study group and control group were 53.16 ± 11.06 and 56.96 ± 12.16 years respectively. In the study group, six patients out of 37, and 20 patients out of 50 in the control group, developed CIL. They received 38 and 105 vials of filgrastim respectively. After excluding those cycles in study group patients, where green jackfruit flour was not taken, the mean number of cycles in which CIL developed (*p* = 0.00) and number of vials of filgrastim taken per cycle (*p* = 0.00) were significantly different from control group and no patient in the study group developed CIL. Inclusion of green jackfruit flour as a dietary intervention prevents chemotherapy-induced leukopenia in patients undergoing chemotherapy along with pegfilgrastim.

## 1. Introduction

Dietary modification with more of vitamins, proteins and fibers for patients receiving chemotherapy has been in regular practice to aid better digestion and adequate nutrition. The intake of food substance as a dietary intervention to function as a hematopoietic growth factor preventing chemotherapy-induced leukopenia (CIL) is a novel concept in oncology [1]. Adjuvant, neo adjuvant or palliative chemotherapy is an essential part of solid tumor management. In addition to the destruction of the tumor cells, cytotoxic chemotherapy suppresses the hematopoietic system and results in impaired host protective mechanisms. Leukopenia is the most common side effect caused by chemotherapy, when white blood cells count drops below 4000. This often leading to limit doses of chemotherapy that can be tolerated [2]. It also causes delay in scheduling subsequent chemo cycles and associated with increased morbidity, mortality, and increased treatment costs [3]. Leukopenia may result in febrile neutropenia (FN), which needs hospitalization for evaluation and administration of proper broad-spectrum antibiotics. Such complications usually result in reductions in dose or delay in treatment, which may compromise clinical outcomes [4,5,6,7,8]. Leukopenia is also associated with exacerbations of other adverse effects of chemotherapy [2], like diarrhea and mucosal ulcers, and respiratory complications like pneumonia, fatigue, electrolytic imbalance, etc. The predictors of CIL complications include advanced age, poor performance status, co-morbidities, and low baseline blood cell counts and high chemotherapy dose intensity [3].

The costs incurred in managing CIL include both direct medical costs and indirect costs following hospital admissions or intensive care that are borne by the patient. Numerous developments have occurred in the management of leukopenia, which include availability of hematopoietic growth factors and improvement in antibiotic therapy, yet this complication remains a central concern in the delivery of cancer chemotherapy [2]. Patients with febrile neutropenia may also develop sepsis, pneumonia or foci of infection, such as cellulitis. Approximately 10% of patients admitted with FN develop serious complications leading to death [9].

This can be mitigated by reducing chemotherapeutical doses or extending dosing intervals. However, these measures reduce the relative dose intensity (RDI) of chemotherapy, which has the disadvantage of reduced survival rates.

The prophylactic use of colony-stimulating factors (CSFs) like pegfilgrastim can reduce the risk, severity, and duration of both severe and febrile neutropenia [10,11,12,13,14,15]. Repeated admissions to manage leukopenia also escalate cost and reduce quality of life. Subcutaneous administration of pegfilgrastim after 24 h of chemotherapy is the current practice rather than daily administration of filgrastim to prevent leukopenia. However, previous randomized control studies show only 43.6% prevention in incidence of FN suggesting there is an unmet need to prevent CIL [16]. Hence, hemopoietic growth factors like filgrastim, are given once leukopenia sets in to prevent FN and eventual death. 

*Artocarpus heterophyllus* Lam., which is commonly known as ‘jackfruit’ is a tropical climacteric fruit native to Western Ghats of India and common in Asia, Africa, and some regions in South America. Jackfruit is rich in nutrients including carbohydrates, proteins, vitamins, minerals, and phytochemicals. Several parts of jackfruit tree including fruits, leaves, and bark have been extensively used in traditional medicine due to its anticarcinogenic, antimicrobial, antifungal, anti-inflammatory, wound healing, and hypoglycemic effects [17]. The tradition of consuming mature green jackfruit as a meal has emerged as an effective alternate to rice to control blood sugar levels due to its soluble fiber, pectin, and lower glycemic load [18]. Researchers have reported that jackfruit seed extract has anti-inflammatory properties indicating its anti-cancer properties [19]. 

Studies [20,21,22] have shown that jackfruit contains many classes of phytochemicals such as carotenoids, which have beneficial effects on several chronic degenerative diseases and cancers. Phytonutrients such as lignans, isoflavones, and saponins in jackfruit contribute to its anticancer property. They may aid in preventing carcinogens by changing milieu and fight against stomach ulcers [23]. The results of a study carried out by Ruiz-Montanez et al. [24] suggested that jackfruit possesses compounds with chemo protective properties to reduce the mutagenicity of aflatoxin B1 (AFB1) and proliferation of cancer cells. In addition to these anticancer effects, researchers have reported that dietary pectin, present in green jackfruit flour, increases survival of bone marrow cells and intestinal crypt stem cells during chemo and radiotherapy. This study was conducted to test whether inclusion of green jackfruit flour in the diet of patients undergoing chemotherapy reduces development of leukopenia. 

## 2. Materials and Methods

This was a retrospective cohort study conducted among patients undergoing chemotherapy for solid tumors at the Department of Oncology, Renai Medicity Hospital, Palarivattom, Cochin, Kerala, India.

The study group comprised of 50 consecutive subjects, who followed the inclusion of green jackfruit flour in their regular diet (referred as JFD) and followed up. The control group comprised of 50 subjects with no specific diet supplements and followed their regular diet. Regular diet of patients in both groups include vegetarian and non-vegetarian dishes with importance to fish accompanied with rice meal or meals prepared with rice or wheat flour. The two groups of patients matched with regards to gender, age, types of solid tumors, types of chemotherapy drugs used, pegylated filgrastim used and chemotherapy protocols followed in the oncology clinic. 

The solid tumors included cancers of the breasts, lungs, buccal mucosa, tongue, pharynx, cancers of head and neck, stomach, esophagus, gastro esophageal junction, colon, rectum, pancreas, cancers of gastro intestinal system, bladder, ovaries, uterus, cervix, and soft tissue sarcomas (Appendix A). The chemotherapeutic drugs used include, taxanes, anthracyclines, holoxan, platinums, targeted chemotherapeutic drugs like herceptin, bevacizumab, nimotuzumab (Appendix A). Chemotherapy was given in the neo adjuvant, adjuvant or palliative indications if performance status was acceptable. Those who received less than three cycles were excluded from either arm.

The study was approved on September 6th, 2019 by the Renai Medicity Hospital institutional ethics committee (Reg. No. ECR/1113/Inst/KL/2018. Ref No.HR/RIMS/158/2019).

### 2.1. Intervention

Apart from the regular diet patients in the study arm were advised to include 30 g of green jackfruit flour to rice or wheat flour they used to prepare meals in equal divided doses with breakfast and dinner, without changing any other elements of day today food habits. *Jackfruit365*™ Green Jackfruit Flour is a patent pending formulation of dehydrated mature green jackfruit fruits fortified with protein and pectin from other parts of jackfruit. 30 g of green jackfruit flour is obtained from 150 g of fresh jackfruit traditionally used to prepare one portion of jackfruit meal for one person and contains 108 Kcal of energy, 20 g of net carbohydrates, 1 g of sugar, 4 g fiber of which 1 g is soluble fiber pectin, 0.4 g of fat, 2 g of protein with 17 types of amino acids, and 0.4 g potassium.

Study group patients given coupons to buy green jackfruit flour packs from the local grocery shops and records were maintained. Patients were advised to take 30 g of green jackfruit flour per day along with staple food from first day of chemotherapy through the last day of each cycle of 21 days, till the completion of the course. Blood examination for Hb, WBC count and platelet counts were done on 5th, 7th, 9th, and 12th day as per standard protocol to watch out for any signs of leukopenia. Patients presented with WBC count of less than 4000 got daily doses of filgrastim till their WBC count reached above 4000. Data of filgrastim use was maintained in hospital patient records and in the pharmacy. No extra invasive or non-invasive tests or procedures were conducted on the patient for the study.

In the study arm addition of Jackfruit flour to the diet was the patient’s choice however the benefits of the JFD were explained to all. The final pathology report was critically evaluated in all the subjects who received neo adjuvant chemotherapy to assess any adverse or beneficial effects of JFD on the tumor impacting chemo response. 

Adoption for JFD was high as it was part of the traditional diet with no known adverse effects. Those who refused JFD initially, in the prospective group, were studied separately. 

### 2.2. Statistical Analysis

Data were checked for completeness and consistency and entered in SPSS version 20 software (IBM, Armonk, NY, US). Descriptive statistics like percentages and mean were used. Analysis was done using ANOVA test for comparison between study and control group. *p* value less than 0.05 was considered statistically significant.

## 3. Results

After excluding patients who received less than three cycles of chemotherapy, analysis was done for a total of 87 patients with 37 patients in study group and 50 patients in control group. Table 1 shows that about 70.3% of the participants were females in study group and 74.0% of the participants were females in control group. About 62.2% and 50% of the participants received neoadjuvant chemotherapy in study group and control group respectively. The mean age of participants in the study group and the control group were 53.16 ± 11.06 years and 56.96 ± 12.16 years respectively (Table 1). 

The participants in control group developed leukopenia in 37 out of 263 cycles of chemotherapy received and 14 out of 232 in the study group. We found six patients in the study group and 20 patients in the control group developed leukopenia respectively. Twenty patients in the control group required 105 vials of filgrastim whereas only six patients in the study group required a total of 38 vials of filgrastim (Table 2).

After excluding those cycles in study group patients, where JFD was not taken, none of the patients in study group developed leukopenia in any cycle out of the 211 cycles of chemotherapy (Table 3).

There was no significant difference in the mean number of cycles of chemotherapy received by patients between the two groups, study group (6.27 ± 2.58) vs control group (5.26 ± 2.52). Significant difference was not found (*p* = 0.15) between the mean number of vials of filgrastim taken per patient in study group (1.02 ± 3.25) and control group (2.10 ± 3.59) respectively (Table 4).

After excluding the cycles in study group in which JFD was not taken, the mean number of cycles of chemotherapy received by patients in the study group was 5.70 ± 2.73 and it was not significantly different (*p* = 0.44) from the patients in the control group (5.26 ± 2.52). In addition, mean number of cycles in which leukopenia occurred in the study group (0.00 ± 0.00, *p* = 0.00) and mean number of vials of filgrastim taken per patient in the study group (0.00 ± 0.00, *p* = 0.00) were significantly lesser as compared to the control group (Table 5).

In the study group, two patients could not take any oral food due to their cancer and could not take JFD in any cycles. Four patients switched to JFD only after they experienced leukopenia without JFD. Patients in study group who followed JFD in all cycles did not experience leukopenia. Green jackfruit flour supplement during chemotherapy, after prophylactic pegfilgrastim, was associated with complete prevention of leukopenia, enhancing the clinical utility of pegfilgrastim (Figure 1).

This was also shown in Figure 2 where monthly average sales of filgrastim from hospital pharmacy reduced from 14.87 vials to just 2.66 vials after the introduction of JFD and it was only required for patients who didn’t take JFD. 

Around fifty percent of the subjects in either arm received neoadjuvant chemotherapy. Pathology reports showed that response to chemotherapy was better with more complete pathological responses in the study arm suggesting absence of adverse response of JFD to the actions of the drugs on the cancer.

## 4. Discussion

Dietary intervention to prevent leukopenia and neutropenia, for those who undergo chemotherapy is reported for the first time. Jackfruit is a tropical fruit available in several countries. The anti-diabetic effect of green jackfruit is well established. The treatment cost reduced by avoiding administration of hematopoietic growth factor, antibiotics, and repeated hospital admissions, could be phenomenal when widely used. The quality of life and psychological satisfaction when leukopenia and neutropenia are reduced during chemotherapy, adds value to this novel dietary intervention. The leukopenia prevention effect of green jackfruit flour is proved in this study by the fact that the mean number of cycles in which leukopenia developed and the number of vials of filgrastim required is significantly lower in the patients who received green jackfruit flour diet. The mechanism of leukopenia prevention effect of jackfruit can be attributed to the pectin component present in green jackfruit flour which protects the intestinal crypt stem cells during chemotherapy [25,26,27,28]. Pectin from jackfruit peel could be obtained by means of various mineral acids and organic acids. The monosaccharide composition of pectin includes galacturonic acid, glucose, arabinose, galactose, and rhamnose. Pectins from jackfruit peel serve as natural antioxidants in food production [29,30]. Antioxidants present in jackfruit which prevent cancer could also prevent the leukopenia caused by chemotherapy.

The possible action of bone marrow stimulation by amino acids present in jackfruit similar to filgrastim need to be investigated [31]. This action could be intensified by increased nutrient absorption with green jackfruit flour diet. Previous studies shown that amino acids provide better nutrition in patients undergoing chemotherapy. Most of the normal nutrient absorption happens at the small intestinal villus surface which is damaged after chemotherapy and alters the normal absorptive mechanism. In these conditions amino acids shown to be better nutrients and reduces the gastrointestinal syndrome following chemotherapy [32]. The same mechanism is plausible in case of green jackfruit flour as it contains significant amount of potassium and amino acids which in turn can reduce the gastrointestinal side effects of chemotherapy (Appendix A). The decrease in neutrophil count caused by chemotherapy is ameliorated by dietary supplementation with a multivitamin or vitamin E. There is growing evidence that these dietary supplements can mimic, intensify, or attenuate the effects of chemotherapeutic agents [33,34]. Presence of these vitamins and beneficial phytochemicals such as carotenoids, lignans, isoflavones, and saponins in green jackfruit flour need to be investigated.

The strength of the study is that this is a novel dietary intervention study with a local plant-based food where the patients in the study group are followed up prospectively in a real life clinical setting. There are two limitations for the study. For study group, no extra data from invasive or non-invasive procedures or tests could be collected and for control group, data were collected retrospectively and limited to available hospital patient records.

Irrespective of the type of tumor or the drugs and regimens used, the end result of leukopenia prevention was identical. There are several reports and meta-analysis examining the impact of Capecitabine in combination with standard neoadjuvant and adjuvant chemotherapy regimens in early breast cancer. These suggest improved outcomes in survival in breast cancer patients. All the patients with poor prognostic variables like triple negative, Er negative, HER2 positive tumors, those with axillary node positivity, high Ki67, vascular invasions, lymphatic invasions, perineural invasions and perinodal extensions showed better survival [35].

The biggest limiting factors are the hand foot syndromes and diarrhea for those who are on oral Capecitabine in pharmacological doses. Therefore, dose reductions or avoidance of the drug will be usually advised, which obviously negates the final outcome. In the study group and control group oral Capecitabine was part of the protocol. Among those who are on JFD, no patient experienced these complications, denoting the fact JFD can even improve survival in those high-risk groups.

## 5. Conclusions

This study concludes that addition of green jackfruit flour as a dietary intervention prevents Chemotherapy Induced Leukopenia (CIL). The impact on global scenario will be recruitment of more patients, who due to toxicity, fail to complete ideal treatments as well as those who refuse chemotherapy due to the fear of developing leukopenia, as experienced by peers.

The combined effect of green jackfruit flour in combination with pegfilgrastim, resulting in complete avoidance of leukopenia is a breakthrough in the field of oncology. This is reported for the first time. Since the scientific community is on the lookout for more natural and plant-based products to treat health related issues on account of unexpected adverse reactions from synthetic agents, the current observational study on green jackfruit flour opens new frontiers for interventions.

Complete avoidance of leukopenia, greatly enhances the quality of life of treating doctor and staff as well, since they need not waste precious hours for treating complications of chemotherapy. Advanced researches on more dose intensive chemotherapy or frequent scheduling also could be done, which may improve the patient’s compliance, outcome and physician’s esteem.

Further prospective multicenter randomized controlled trials with adequate sample and additional physical and hematological data to establish evidence of leukopenia prevention effect of green jackfruit flour can be studied in different disease subgroups, with different regimens both in adult and pediatric category. The cost effectiveness in preventing leukopenia when translated into money saved globally, will be in billions.

## Figures and Tables

**Figure 1 biomolecules-10-00218-f001:**
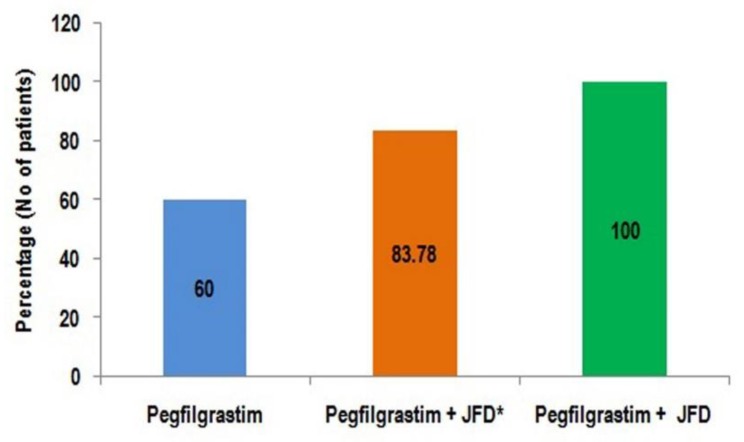
Effect of JFD in the prevention of Leukopenia; * Including six patients who could not take JFD in all cycles.

**Figure 2 biomolecules-10-00218-f002:**
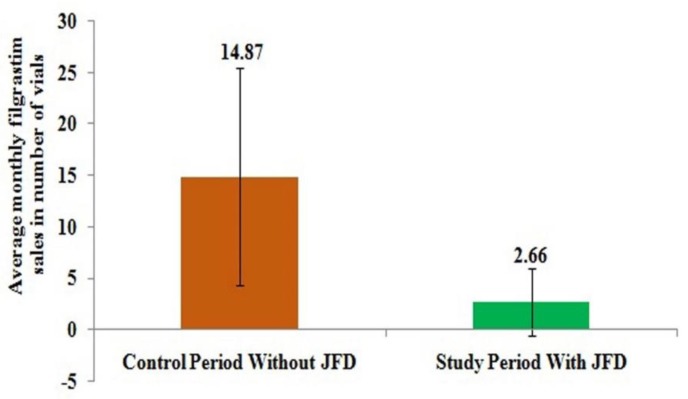
Average monthly sales quantity in vials for filgrastim from hospital pharmacy.

**Table 1 biomolecules-10-00218-t001:** Baseline characteristics of participants.

Variable	Category	Study	Control
Gender	Male	11 (29.7%)	13 (26.0%)
	Female	26 (70.3%)	37 (74.0%)
Therapy given	Adjuvant	12 (32.4%)	21 (42.0%)
	Neo adjuvant	23 (62.2%)	25 (50.0%)
	Palliative	2 (5.4%)	4 (8.0%)
Age (Mean ± SD)		53.16 ± 11.06	56.96 ± 12.16

**Table 2 biomolecules-10-00218-t002:** Descriptive statistics for patients in study and control group.

	Study Group	Control Group
Total number of cycles of chemotherapy given	232	263
Number of cycles in which leukopenia was present	14	37
Number of vials of filgrastim taken	38	105
Number of patients for whom leukopenia occurred	6	20

(Including those cycles from study group patients where jackfruit flour diet (JFD) was not taken).

**Table 3 biomolecules-10-00218-t003:** Descriptive statistics for patients in study and control group.

	Study Group	Control Group
Total number of cycles of chemotherapy given	211	263
Number of cycles in which leukopenia was present	0	37
Number of vials of filgrastim taken	0	105
Number of patients for whom leukopenia occurred	0	20

(After excluding those cycles from study group patients where JFD was not taken).

**Table 4 biomolecules-10-00218-t004:** Difference between parameters in study and control group.

	Study Group (mean ± SD)	Control Group (mean ± SD)	*p*-Value
Number of cycles of chemotherapy per patient	6.27 ± 2.58	5.26 ± 2.52	0.07
Number of vials of filgrastim taken per patient	1.02 ± 3.25	2.10 ± 3.59	0.15

**Table 5 biomolecules-10-00218-t005:** Difference between parameters in study and control group.

	Study Group (Mean ± SD)	Control Group (Mean ± SD)	*p*-Value
Number of cycles of chemotherapy per patient	5.70 ± 2.73	5.26 ± 2.52	0.44
Number of cycles in which leukopenia was present per patient	0	0.74 ± 1.05	0.00
Number of vials of filgrastim taken per patient	0	2.10 ± 3.59	0.00

(After excluding those cycles from study group patients where JFD was not taken).

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
