# Peer review of "Efficacy of Jackfruit365™ Green Jackfruit Flour Fortified Diet on Pegfilgrastim to Prevent Chemotherapy-Induced Leukopenia, Irrespective of Tumor Type or Drugs Used—A Retrospective Study"

_biomolecules, 2020, doi:10.3390/biom10020218_

Round 1

Reviewer 1 Report

The manuscript entitled “Efficacy of Jackfruit365™ Green Jackfruit Flour Fortified Diet on Pegfilgrastim to Prevent Chemotherapy-induced Leukopenia, Irrespective of Tumor Type or Drugs Used - A Retrospective Study” by Varughese et al. is  interestingNevertheless the Authors did not escape some ambiguities in the text. Suggestions of the reviewer are presented below: 

Some information in the Introduction are not directly related to the subject of the manuscript. For example fragment in lines 81-81 shows commonly known information about roles of diet in chemotherapy. The reviewer suggests to remove this fragment.  The Authors should to add the clear explanation why they have performed study on green jackfruit in the introduction.  In the introduction components of green jackfruit flour should be described more precisely. The reviewer suggests to add table with names of components of flour and their potential roles in human health.  The exact name of ethics committee and date of obtaining of the degree should be added.  General information about diet of patients in control and experimental groups (as far as possible in materials and methods should be added  What statistical test has been used (Anova, t-Student or other)? The producer of statistical programme should be added  Units (%) should be added to values in parentheses in table 1  In table 4 in column of control group should be “mean±SD” instead “meanSD  In table 5 in the opinion of reviewer should be “0” instead “0.00± 0.00”  Information about filgrastim in the first paragraph of discussion is not discussion of obtained results and therefore it is more appropriate to introduction  In the opinion of the reviewer the roles and mechanisms of the particular ingredients of green jackfruit flour in observed results should be discussed more precisely. 

In spite of the mentioned above small inadequacies, the reviewer thinks that the article can be published after minor revision . 

Author Response

We thank reviewer1 for the valuable suggestions and made the following corrections: 

For example fragment in lines 81-81 shows commonly known information about roles of diet in chemotherapy. The reviewer suggests removing this fragment.

Response: agree and removed

The Authors should to add the clear explanation why they have performed study on green jackfruit in the introduction.

Response: Yes, evidence on dietary pectin (present in jackfruit) protecting intestinal crypt stem cells during chemo and radiotherapy is mentioned as the main reason to conduct this study. (lines 89-93)

In the introduction components of green jackfruit flour should be described more precisely. The reviewer suggests to add table with names of components of flour and their potential roles in human health.

Response: Green Jackfruit Flour is standardized by its nutritional profile than components. We have added full nutritional profile in methods section for more clarity. Potential roles of soluble fiber pectin is discussed in introduction and in discussion.

The exact name of ethics committee and date of obtaining of the degree should be added. General information about diet of patients in control and experimental groups (as far as possible in materials and methods should be added.

Response: agree, added full name of institution  and date of approval

What statistical test has been used (Anova, t-Student or other)? The producer of statistical programme should be added Units (%) should be added to values in parentheses in table 1 In table 4 in column of control group should be “mean±SD” instead “meanSD” In table 5 in the opinion of reviewer should be “0” instead “0.00± 0.00”

Response: ANOVA has been used to compare between the groups, we have added this now. The analysis has been done using SPSS v 20 already mentioned. All changes suggested for tables have been incorporated in the manuscript under Table 1, 4 and 5.

Information about filgrastim in the first paragraph of discussion is not discussion of obtained results and therefore it is more appropriate to introduction.

Response: Thank you for pointing this out. We discussed and decided it is better to remove this paragraph on filgrastim. Removed

In the opinion of the reviewer the roles and mechanisms of the particular ingredients of green jackfruit flour in observed results should be discussed more precisely.

Response: Thank you we called out the role of pectin and added supplementary data on amino acids. Further analysis on mechanism action would require basic research and was outside scope of this study. We will include this in a future paper.

Reviewer 2 Report

Concern addressed.  

Author Response

We thank Reviewer 2 for the encouraging comments and support

Reviewer 3 Report

The presented manuscript regards leukopenia and the natural jackfruit application for supportive therapy. The intention of the authors was good, but the overall representation of the manuscript must be improved. There are some minor remarks:
-line 89 - double dot, line 152 remove comma
-line 91 - should be chemo- and radiotherapy
-in the introduction section indicate clearly the aim of the study
-authors indicated that jackfruit has numerous classes of components i.e. phytochemical compounds. Could you address any data which of these compounds can have the most impact on its activity in leukopenia?
-there is indicated that "Blood examination for Hb, WBC count and platelet counts were done", do you have a graphical representation of the obtained results, this should be also discussed, there is no reference to these data
-indicate what test was implemented for statistical analysis,
-material and methods should be ordered, there is not clear what was the design of the control and treated group. I can find some data based only on the tables from results.
-line 174-176 - the sentence and zero results are not clear, revise this sentence.
-fig. 1 has bad quality, and the unit should be (%) but of what? , description with the star superscript can be added to fig.
-description, the legend is not necessary if there are only 3 bars with the same description, is there any SD?
-fig 2 also has bad quality, and in the axis' description remove dot
- the reference section contains only one paper from 2019, really is there no fresh research?
- English should be revised, there are some stylistic and punctuation mistakes

Author Response

We thank Reviewer 3 for the detailed review and comments for improvements. 

Below are the changes and responses to comments: 

-line 89 - double dot, line 152 remove comma -line 91 - should be chemo- and radiotherapy

Response: thank you and corrections done.

In the introduction section indicate clearly the aim of the study -authors indicated that jackfruit has numerous classes of components i.e. phytochemical compounds. Could you address any data which of these compounds can have the most impact on its activity in leukopenia?

Response: Yes, dietary pectin was the main lead for us to focus on leukopenia prevention and has been called out as the main aim of the study. Effect of phytochemical compounds are reported for their anticancer properties than prevention of leukopenia.

-there is indicated that "Blood examination for Hb, WBC count and platelet counts were done", do you have a graphical representation of the obtained results, this should be also discussed, there is no reference to these data -indicate what test was implemented for statistical analysis, -material and methods should be ordered, there is not clear what was the design of the control and treated group. I can find some data based only on the tables from results.

Response: Though the examinations for Hb, WBC and platelet counts were conducted for each patient as part of the clinic protocol, in most cases this data is with individual patients and not part of hospital patient records. Study was restricted to hospital records, a limitation already called out in discussion section. Only patients with Leukopenia where WBC count is verified to be below 4000 is given filgrastim at the hospital. 

Test ANOVA was used to compare between the groups, we have added it to manuscript. The analysis was been done using SPSS v 20 and already mentioned

-line 174-176 - the sentence and zero results are not clear, revise this sentence.

Response: Yes changes have been done.

-fig. 1 has bad quality, and the unit should be (%) but of what? , description with the star superscript can be added to fig. -description, the legend is not necessary if there are only 3 bars with the same description, is there any SD?

Response: The quality has been improved by increasing the resolution to 600dpi and descriptions in both axis made more clearer. The % denotes no of patients, corrected. Star superscript was added to the description. The legend removed , there was no SD for fig1.

fig 2 also has bad quality, and in the axis' description remove dot

Response: The quality has been improved by increasing the resolution to 600dpi and x axis and y axis edited for more clarity. The dot was removed.

the reference section contains only one paper from 2019, really is there no fresh research?

Response: We have added one more reference from 2019 on Jackfruit. Jackfruit flour and testing leukopenia prevention are new developments, hence very less research so far. We hope this study results will encourage researchers to conduct more study on jackfruit and jackfruit flour for Leukopenia.

English should be revised, there are some stylistic and punctuation mistakes

Response: we have got a native English speaker to review and corrected those errors.  

This manuscript is a resubmission of an earlier submission. The following is a list of the peer review reports and author responses from that submission.

Round 1

Reviewer 1 Report

I really find this article interesting and it will really be good to find something that can prevent or reduce chemotoxicity and that too from natural sources. I still feel that more data could have been generated to back the current report. Also the explanation in the discussion could be more simplified.

Author Response

Response:

Thank you for your comments, we appreciate your observation on the importance of this study to reduce chemotoxicity using a natural source.

Study Design & Data: We agree a prospective randomized control trial in the same time period would have been a better design. This study was started just as an anecdotal observation in a real life clinical setting with significant limitations on number of patients and resources. The study was conducted without changing any of the existing treatment protocols already followed in the clinic and without conducting any extra invasive or noninvasive tests or procedures on patients. However after observing the unexpected overwhelming results, that too addressing one of the major impediments in cancer treatments like chemotoxicity, with a widely available natural plant based food which used to be part of the local traditional diet, we felt the urgency to inform the scientific world so that researchers based in larger hospitals with better resources could take up better designed prospective studies on larger patient population. Hence we took approval from hospital ethics committee to use retrospective data from hospital records to publish the current study, than waiting for an indefinite period for a new prospective study to be completed, that too with significant limitation on resources.

We will edit the discussion section to simplify it further.

Reviewer 2 Report

Major comments: The authors examined the effect of jackfruit flour diet (JFD) in patients receiving chemotherapeutic treatment against solid tumor. The major concern is the relative lack of impact and salience. Several important points stated below need attention. (1) How about the adverse side effects caused by the intake of JFD alone? This point should be addressed. (2) How about the dose effect of JFD on the treatment of patients? (3) How do the authors think about the mechanism of action of JFD for the prevention of chemotoxicity in the current study? (4) It is of interest what the potential active component(s) of JFD is. It would be desirable for the authors to discuss this point.   (5) How did the authors evaluate the synergistic effect of JFD in the current study? The method and the results should be addressed. (6) What about the importance or significance of Fig. 2? This point should be explained. (7) Is there any possibility that the dietary constituents of the patients affect the preventive effect of chemotoxicity by JFD intake?

Author Response

We thank for the detailed feedback and list of questions. Please see our responses to the list questions below. Our comments on the overall study design is provided at the end. 

How about the adverse side effects caused by the intake of JFD alone? This point should be addressed.

Response: JFD is a natural plant based food and used to be part of the local traditional diet like a Mediterranean salad, with no known adverse effect. We will address this further in the materials and method section for more clarity.

How about the dose effect of JFD on the treatment of patients?

Response: In traditional diet, 150 grams of green jackfruit is used to prepare the jackfruit meal once a day to replace rice meal. 150 grams of green jack fruit is equivalent to 30g of JFD in dehydrated form. So in our study patients were advised to take 30 grams of JFD per day in line with the local tradition.

How do the authors think about the mechanism of action unless otherwise stated of JFD for the prevention of chemotoxicity in the current study?

Response: The mechanism of action we considered at the time of introducing JFD was the role of dietary pectin in chemo and radiotoxicity prevention. Subsequently we learned the availability of 17 amino acids in JFD as another plausible mechanism for chemotoxicity prevention and repair. Both are mentioned in discussions.    

It is of interest what the potential active component(s) of JFD is. It would be desirable for the authors to discuss this point.

Response: Based on existing literature review we found the chemotoxicity prevention of JFD is mainly due to the presence of Pectin and amino acids which is included in discussion as plausible mechanism of action. However, the role of other active components in JFD will need to be considered in future basic research studies.

How did the authors evaluate the synergistic effect of JFD in the current study?

Response: The study was conducted without altering existing chemotherapy protocols followed in the clinic, apart from the addition of 30g JFD per day along with their regular food. Prophylactic administration pegfilgrastim is used in the clinic to prevent chemotoxicity. In spite of using pegfilgrastim, 40% patients experienced chemotoxicity in the control period without JFD. We could study the effect of Pegfilgrastim alone and Pegfilgrastim with JFD. However, we admit existing protocol at the clinic didn’t allow us to evaluate the effect of JFD alone without pegfilgrastim and will be included in a future basic research. If reviewer feels the use of the word ‘synergistic’ in title is not accurate, we can replace ‘synergistic effect’ with ‘combined effect’.      

What about the importance or significance of Fig. 2? This point should be explained.

Response: Fig 2 clearly shows there was a significant (80%) drop in demand for filgrastim from the hospital pharmacy between control and study period. This is a secondary evidence that incidents of w.b.c. count falling below 4000 due to chemotoxicity was reduced drastically after JFD was introduced. We feel this is an important piece of evidence for this study. To explain this point clearer, we will replace the current monthly demand graph with a new graph showing monthly average demand for filgrastim without JFD (14.87 vials) and with JFD (2.66 vials) and provide more explanation.

Is there any possibility that the dietary constituents of the patients affect the preventive effect of chemotoxicity by JFD intake?

Response: Only variable in the diet without JFD and with JFD was addition of 30g JFD per day along with their regular meals. Other than an equivalent volume reduction in rice or wheat flour, no other dietary constituents got altered. Hence we don’t think any other dietary constituent could have caused the preventive effect of chemotoxicity other than JFD.

Study Design & Data: We agree a prospective randomized control trial in the same time period would have been a better design. This study was started just as an anecdotal observation in a real life clinical setting with significant limitations on number of patients and resources. The study was conducted without changing any of the existing treatment protocols already followed in the clinic and without conducting any extra invasive or noninvasive tests or procedures on patients. However after observing the unexpected overwhelming results, that too addressing one of the major impediments in cancer treatments like chemotoxicity, with a widely available natural plant based food which used to be part of the local traditional diet, we felt the urgency to inform the scientific world so that researchers based in larger hospitals with better resources could take up better designed prospective studies on larger patient population. Hence we took approval from hospital ethics committee to use retrospective data from hospital records to publish the current study, than waiting for an indefinite period for a new prospective study to be completed, that too with significant limitation on resources.

Reviewer 3 Report

The manuscript entitled "Synergistic effect of Jackfruit365™ green jackfruit flour fortified diet on pegfilgastrim to prevent chemotoxicity, irrespective of tumor type or drugs used - a retrospective study" by Thomas Varughese and James Joseph has evaluated the effect of Jackfruit derived dietary flour on patients for chemotherapy induced chemotoxicity prevention.

While the study is novel and the study design may be acceptable to an extent the data needs improvement. The control group patients should be evaluated during the same study period receiving the similar chemotherapy drugs under same/similar conditions without the experimental dietary supplement Jackfruit365. However since the study has been completed the current group of patients as control subjects is acceptable.

However, the major concern is the missing medical data of toxicity evaluation in the pre-treatment, post treatment and follow-up of patients during the chemotherapy and dietary supplement substituted treatment in the control and experimental patient groups.

Inclusion of these medical data in the study will improve the quality of the manuscript as no other medical evidence is available for the claims that the Jackfruit36 prevents chemotoxicity.

Author Response

However, the major concern is the missing medical data of toxicity evaluation in the pre-treatment, post treatment and follow-up of patients during the chemotherapy and dietary supplement substituted treatment in the control and experimental patient groups

Response: Thank you for your comments, understanding of our limitations and suggestions.

Since JFD was based on a traditional diet followed by the local population for centuries and using a natural plant based food, like any other dried fruits and vegetables commonly used, with no known toxicity, we didn’t feel the need to conduct toxicity evaluation at the time of introducing JFD. In the present study, we observed the effect of JFD in the prevention of chemotoxicity and also we have not experienced any adverse effects, other than complaints about minor alteration in taste of food prepared with JFD. This implies JFD might be safe for a wider population. However, in our study we have only observed short outcomes and we should consider the long term effect of JFD to authenticate its safety in future prospective studies, especially for population alien to Jackfruit meal.

Study Design & Data:

We agree a prospective randomized control trial in the same time period would have been a better design. This study was started just as an anecdotal observation in a real life clinical setting with significant limitations on number of patients and resources. The study was conducted without changing any of the existing treatment protocols already followed in the clinic and without conducting any extra invasive or noninvasive tests or procedures on patients. However after observing the unexpected overwhelming results, that too addressing one of the major impediments in cancer treatments like chemotoxicity, with a widely available natural plant based food which used to be part of the local traditional diet, we felt the urgency to inform the scientific world so that researchers based in larger hospitals with better resources could take up better designed prospective studies on larger patient population. Hence we took approval from hospital ethics committee to use retrospective data from hospital records to publish the current study, than waiting for an indefinite period for a new prospective study to be completed, that too with significant limitation on resources.

Round 2

Reviewer 2 Report

Concern addressed.

Reviewer 3 Report

The manuscript has not provided the clinical chemotoxicity data/patient for the study. Since the study is based on retrospective study protocol, the validity of the study can be only achieved if proper clinical data is presented along with the statistical analysis. Clinical data is also relevant for the study as the study encompasses subjects irrespective of tumor types.